# Transcriptomic analysis reveals reduced transcriptional activity in the malaria parasite *Plasmodium cynomolgi* during progression into dormancy

Nicole L Bertschi[1†], Annemarie Voorberg-van der Wel[2†], Anne-Marie Zeeman[2], Sven Schuierer[1], Florian Nigsch[1], Walter Carbone[1], Judith Knehr[1], Devendra K Gupta[3], Sam O Hofman[2], Nicole van der Werff[2], Ivonne Nieuwenhuis[2], Els Klooster[2], Bart W Faber[2], Erika L Flannery[3], Sebastian A Mikolajczak[3], Vorada Chuenchob[3], Binesh Shrestha[1], Martin Beibel[1], Tewis Bouwmeester[1], Niwat Kangwanrangsan[4], Jetsumon Sattabongkot[5], Thierry T Diagana[3], Clemens HM Kocken[2]*, Guglielmo Roma[1]*

[1]Novartis Institutes for BioMedical Research, Novartis Pharma AG, Basel, Europe; [2]Department of Parasitology, Biomedical Primate Research Centre, Rijswijk, The Netherlands; [3]Novartis Institute for Tropical Diseases, Novartis Pharma AG, Emeryville, United States; [4]Department of Pathobiology, Faculty of Science, Mahidol University, Bangkok, Thailand; [5]Mahidol Vivax Research Unit, Faculty of Tropical Medicine, Mahidol University, Bangkok, Thailand

**\*For correspondence:**
kocken@bprc.nl (CHMK);
guglielmo.roma@novartis.com (GR)

[†]These authors contributed equally to this work

**Abstract** Relapses of *Plasmodium* dormant liver hypnozoites compromise malaria eradication efforts. New radical cure drugs are urgently needed, yet the vast gap in knowledge of hypnozoite biology impedes drug discovery. We previously unraveled the transcriptome of 6 to 7 day-old *P. cynomolgi* liver stages, highlighting pathways associated with hypnozoite dormancy (Voorberg-van der Wel et al., 2017). We now extend these findings by transcriptome profiling of 9 to 10 day-old liver stage parasites, thus revealing for the first time the maturation of the dormant stage over time. Although progression of dormancy leads to a 10-fold decrease in transcription and expression of only 840 genes, including genes associated with housekeeping functions, we show that pathways involved in quiescence, energy metabolism and maintenance of genome integrity remain the prevalent pathways active in mature hypnozoites.
DOI: https://doi.org/10.7554/eLife.41081.001

## Introduction

*Plasmodium vivax* malaria puts 35% of the world's population at risk of disease (*World Health Organization, 1987*). A large barrier to *P. vivax* eradication is afforded by the parasite's ability to cause relapsing disease weeks to months or even years after the primary mosquito-mediated infection (*White, 2011*). This aspect of *P. vivax* infection is caused by a dormant liver stage form of the parasite, named the hypnozoite. Hypnozoites can reactivate through unknown mechanisms, continue development into liver schizonts and cause renewed disease. While prophylactic drugs can prevent initial parasite development in the liver (*Zeeman et al., 2016*), there is a need for new radical cure compounds targeting established hypnozoites. There is a vast gap in knowledge surrounding hypnozoite biology that causes a significant setback in developing a novel radical cure drug that can eliminate hypnozoites. Such a drug is urgently needed to aid in malaria eradication because the current radical curative drugs cannot be used in all persons in endemic areas. For instance, primaquine and

the recently FDA-approved tafenoquine cannot be administered to patients with glucose-6-phosphate dehydrogenase (G6PD) deficiency, a common genetic disorder in malaria endemic countries, due to serious adverse side-effects and life-threatening drug-induced hemolysis (*Mazier et al., 2009*; *Wells et al., 2010*). For this reason, new and better drugs are urgently needed. One compound that appeared promising in *in vitro* screens, a phosphatidylinositol 4-kinase (PI4K) inhibitor (*Zeeman et al., 2014*), was shown to be active against hypnozoites in a prophylactic, but not radical cure dosing scheme (*Zeeman et al., 2016*). This indicates that 'early' hypnozoites differ from 'established' hypnozoites and suggests differences in active cellular pathways between the two forms. *P. vivax* hypnozoite biology research is hampered by the absence of an *in vitro* blood stage culture system, which makes the development of *P. vivax* research tools dependent on patient material as a source for sporozoites. The *P. cynomolgi* simian malaria, closely related to *P. vivax*, is a well-validated model for human *P. vivax* malaria useful to study the disease relapse caused by the reactivation of liver hypnozoites. *P. cynomolgi in vitro* liver stage cultures have been established and in combination with the established transfection technology for this parasite this provides unique opportunities for studies into hypnozoite biology. To investigate differences between different aged hypnozoites, we used genetically engineered fluorescent *P. cynomolgi* parasites (*Voorberg-van der Wel et al., 2013*), *in vitro P. cynomolgi* liver stage culture (*Zeeman et al., 2014*), cell-sorting and RNA-seq to expand the recently published liver stage transcriptomes (6 to 7 day-old *in vitro* cultured hypnozoites and replicating liver stages) (*Voorberg-van der Wel et al., 2017*) by transcriptionally profiling malaria parasite liver stages after 9 to 10 days of *in vitro* culture. The transcriptomic analysis we describe here, together with the previously published study show that hypnozoite maturation is accompanied by a 10-fold decrease in transcriptional activity and the expression of 840 genes of which only ~4% at higher levels ($\geq$10 FPKM). Although established hypnozoites continue to express a subset of genes, a marker gene which specifically distinguishes hypnozoites from schizonts was not detected. Genes and pathways associated with quiescence, energy metabolism and maintenance of genome integrity remain prevalently expressed in mature hypnozoites, indicating that the hypnozoite stage may be defined by a network of regulatory factors cooperating in the maintenance of genome stability and in the epigenetic control of gene expression.

## Results

### Transcriptome analysis of malaria parasite liver stages after 9 to 10 days of *in vitro* culture

To gain further understanding of dormancy mechanisms described in hypnozoites at days 6 and 7, we FACS-purified hepatocytes containing GFP-expressing hypnozoites (GFPlow) and liver schizonts (GFPhigh) at later time points, 9 and 10 days after *P. cynomolgi* M strain sporozoite infection. Mean fluorescence intensities, a measure that depends on cell size and fluorescence intensity, slightly increased from day 6 to day 9 in the GFPlow samples, illustrating a small increase in the volume of hypnozoites over time (*Mikolajczak et al., 2015*). A more pronounced increase over time was observed in the GFPhigh samples, indicating significant parasite growth (*Figure 1—figure supplement 1A*). Microscopic analysis of quality control samples taken from the FACS-sorted parasites revealed advanced schizogony at day 9 (GFPhigh samples) and small forms that were, similar to day 6 sorted forms, in- or outside hepatocytes (GFPlow samples) (*Figure 1—figure supplement 1B*). The GFPlow samples contained substantial amounts of uninfected hepatocytes, possibly due to increased hepatocyte autofluorescence as a result of prolonged culture.

RNA-sequencing resulted in a dataset containing two independent schizont and hypnozoite samples at day 9, and four independent schizont and hypnozoite samples at day 10 (*Supplementary file 1*). Gene expression data showed high concordance between biological replicates (*Figure 1—figure supplement 2*). We observed a strong correlation between schizont ($\rho$ = 0.99 at day 9 and $\rho$ = 0.97 at day 10) and hypnozoite transcriptomes ($\rho$ = 0.85 at day 9 and $\rho$ = 0.87 at day 10), however the hypnozoite samples did not correlate as highly as schizonts between the time points ($\rho$ = 0.82 vs $\rho$ = 0.97). While schizonts revealed a large consensus between genes expressed at days 9 and 10, day 9 and day 10 hypnozoites showed little overlap because a significant number of genes showed

low to no transcription levels at day 9 and high expression at day 10 (*Figure 1—figure supplement 3A*). These included genes involved in merozoite maturation (for example MSP1, AMA1 and SUB1) (*Supplementary file 2*) (*Figure 1—figure supplement 3B*). In contrast, housekeeping genes (e.g. hsp70) showed similar expression levels at these time points (*Figure 1—figure supplement 3B*). Given the presence of transcripts of genes involved in schizont maturation and merozoite invasion, we hypothesized and confirmed by RNA fluorescence in situ hybridization (FISH) a contamination of day 10 hypnozoites with schizonts and/or released merozoites. FISH with a probe against *msp1* was positive for schizonts at day 10, while no transcripts were detected in hypnozoites at day 10 (*Figure 1—figure supplement 3C*). In contrast, transcripts for *hsp70* were found both in schizonts and hypnozoites, except in fully mature schizonts (*Figure 1—figure supplement 3C*).

These results show that presence of schizont transcripts in the day 10 GFPlow samples requires careful *in vitro* validation for each transcript that is detected in hypnozoites. For this reason, we decided to exclude the day 10 timepoint from further analyses.

## Late stage liver parasites reveal features of advanced schizogony and dormancy

We found that later stage (day 9) liver schizonts express a similar number of genes as those observed at days 6/7 (5640 genes at day 9 vs 5702 at day 6/7, with FPKM $\geq$1). However at day 9 genes are expressed at higher levels (5138 genes at day 9 vs 3582 at day 6/7, with FPKM $\geq$10) (*Figure 1*). Most of the genes (94%, 3380 out of total 3582) showing already high expression level at day 6/7 ($\geq$10 FPKM) remain highly expressed at day 9. However, at day 9 we find an additional 1758 genes with increased expression levels ($\geq$10 FPKM) compared to day 6/7. These include genes that are involved in merozoite formation (PALM) or required by merozoites for red blood cell invasion

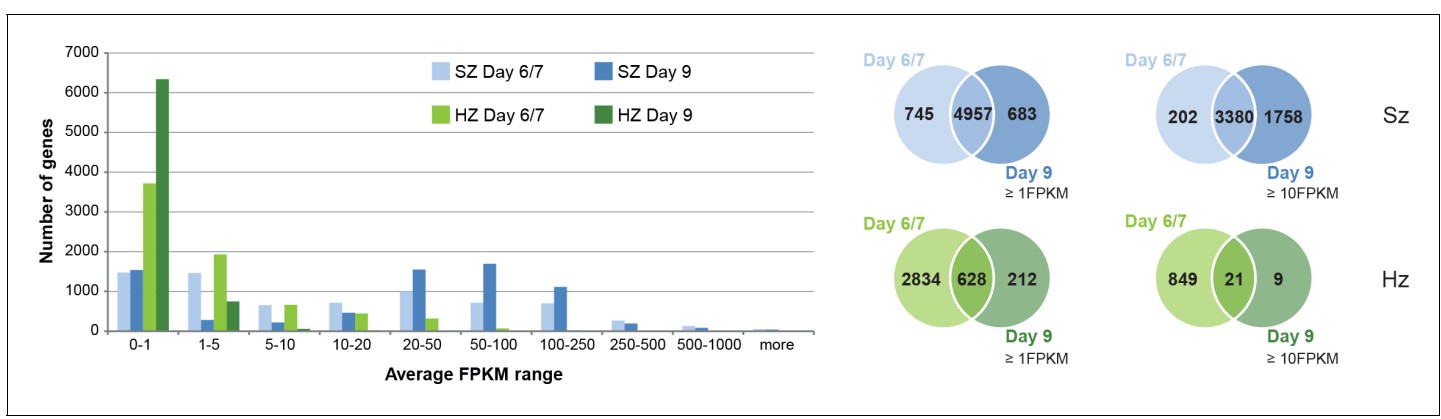

**Figure 1.** Comparison of transcriptional activity in schizonts (SZ) and hypnozoites (HZ) at day 6/7 and day 9. Left panel: Distribution of average gene expression values in the schizont samples at day 6/7 (light blue; n = 3) and at day 9 (dark blue; n = 2), and distribution of average gene expression values in the hypnozoites samples at day 6/7 (light green; n = 4) and at day 9 (dark green; n = 2). FPKM, Fragments per kilobase of transcript per million mapped reads. Upper right panel: Venn diagrams show the overlap of genes expressed $\geq$1 FPKM and $\geq$10 FPKM, respectively, at day 6/7 and at day 9 in the schizont samples. Lower right panel: Venn diagrams show the overlap of genes expressed $\geq$1 FPKM and $\geq$10 FPKM, respectively, at day 6/7 and at day 9 in the hypnozoite samples.

DOI: https://doi.org/10.7554/eLife.41081.002

The following figure supplements are available for figure 1:

**Figure supplement 1.** Quality check of FACS-sorted parasite cells.
DOI: https://doi.org/10.7554/eLife.41081.003

**Figure supplement 2.** Correlation of gene expression values between biological replicates.
DOI: https://doi.org/10.7554/eLife.41081.004

**Figure supplement 3.** Possible contamination of day 10 hypnozoites with schizont derived transcripts.
DOI: https://doi.org/10.7554/eLife.41081.005

**Figure supplement 4.** Diagram showing mean gene expression values (FPKM) of schizogony markers in the schizont samples at day 6/7 and day 9.
DOI: https://doi.org/10.7554/eLife.41081.006

**Figure supplement 5.** Expression levels of specific genes of interest.
DOI: https://doi.org/10.7554/eLife.41081.007

(RhopH3, AMA1, MSP1) and that are already reported to be expressed in late stage liver schizogony (*Sherling et al., 2017*; *Haussig et al., 2011*). Moreover, the sexual stage-specific marker Pvs16, which was previously shown to be expressed in late stage *P. vivax* liver schizonts, also shows upregulation at day 9 compared to day 6/7 (*Figure 1—figure supplement 4*) (*Roth et al., 2018*). In addition to invasion pathways, cytoskeleton motor protein pathways showed increased transcriptional activity at day 9 (97%) compared to day 6 (69%) (*Supplementary file 4*). These actin-myosin motor proteins play crucial roles in apicomplexan host cell invasion (*Bargieri et al., 2014*). It is well known in blood stages that these genes exhibit peak expression in late schizont stage only, which correlates with schizont maturation (*Bozdech et al., 2003*). The high transcription level of these motor protein genes in day 9 liver schizonts suggests near to complete schizont maturation at this time point. Overall, the increased transcriptional activity observed in the later stage liver schizonts reveals molecular events associated with progressing schizogony.

In contrast, day 9 hypnozoites showed a lower transcriptional activity compared to day 6/7 with only few genes expressed at higher levels (30 genes at day 9 vs 870 at day 6/7, with FPKM $\geq$10). This is also reflected in the significant 10-fold decrease of gene expression levels in hypnozoites (e.g. from 5.8 avg FPKM at day 6/7 down to 0.6 avg FPKM at day 9: Wilcoxon test p-value<0.02857). We observed that only 18% of genes expressed in days 6/7 hypnozoites (628 out of 3462 genes with FPKM $\geq$1) are as well expressed at day 9 (628 out of 840 genes with FPKM $\geq$1). This transcriptional decrease is even more pronounced for highly expressed genes where only 2.4% of genes expressed at day 6/7 (21 out of 870 genes with FPKM $\geq$10) show high transcriptional activity also at day 9 (21 out of 30 genes with FPKM $\geq$10) (*Figure 1*). The small number of highly expressed genes at day 9 is most likely the result of an overall reduced transcriptional activity ongoing during hypnozoite maturation. Indeed, the nine genes found to be highly expressed only in day 9 hypnozoites mostly encode ribosomal and histone proteins which are also expressed at day 6/7 but just below the cutoff of 10 FPKM. In contrast, over 70% of all genes expressed $\geq$10 FPKM at day 6/7 are expressed below 1 FPKM at day 9 (*Supplementary file 3*). These results provide for the first time a strong evidence that maturation of the dormant liver stage is associated with continued reduction of its transcriptional activity.

In the published dataset of day 6/7 a few hypnozoite-specific genes were identified with significantly higher expression levels compared to schizonts and could thus be hypnozoite markers (*Voorberg-van der Wel et al., 2017*). However, all these genes show expression levels <1 FPKM in hypnozoites at day 9, and higher transcriptional levels in schizonts (*Figure 1—figure supplement 5A*). We hence do not find any gene that is exclusively expressed in the hypnozoite population at day 9, concluding that there is no specific marker for dormant stages at the transcriptional level. Notably, the liver-stage-specific AP2-L protein, which is a member of the plant-derived Apicomplexan Apetala2 (ApiAP2) family of transcription factors, showed equivalent transcription levels in hypnozoites at day 6/7 and day 9 (*Balaji et al., 2005*; *Iwanaga et al., 2012*). In contrast, AP2-Q which has been proposed as a master regulator of transcription in hypnozoites (*Cubi et al., 2017*), showed only very low transcription levels in hypnozoites at day 9 (*Figure 1—figure supplement 5B*), which corroborates the results of the recently published *P. vivax* hypnozoite transcriptome (*Cubi et al., 2017*; *Gural et al., 2018*). Interestingly, in the *P. vivax* transcriptome, Gural et al. observed another ApiAP2 transcription factor (PV01_0916300) with high transcript abundance in *P. vivax* hypnozoites (*Gural et al., 2018*). Indeed, we also found transcription of this AP2-encoding gene (PcyM_0918000) in *P. cynomolgi* hypnozoites at day 6/7 and day 9 (*Figure 1—figure supplement 5B*).

To evaluate the potential to apply insights from the transcriptome data for drug discovery, we looked at the expression levels of clinically and chemically validated drug targets (*Voorberg-van der Wel et al., 2017*). For almost all drug targets transcription levels had dropped to low levels in day 9 hypnozoites (*Figure 1—figure supplement 5B*). Transcription levels of PI4K in hypnozoites at days 6/7 and day 9 are almost zero, which is in line with the proposed early mode of action that explains the lack of radical cure activity of the drug despite its strong prophylactic activity (*Figure 1—figure supplement 5B*) (*Zeeman et al., 2016*) and which aligns with the *P. vivax* hypnozoite data of *Gural et al., 2018*. In contrast, day 9 hypnozoites show still a significant number of transcripts for eukaryotic elongation factor 2 (eEF2), albeit at lower levels than at days 6/7, warranting further research into this target for potential radical cure.

Taken together, our data suggest that while late schizogony at day 9 is associated with increased transcriptional activity compared to days 6/7, continued dormancy at day 9 is associated with a decrease in gene transcription.

## Dormancy in maturing hypnozoites is associated with a general metabolic shutdown

We previously reported that schizonts at days 6/7 express over 90% of the malaria pathways annotated in PlasmoDB (*Voorberg-van der Wel et al., 2017*). These included energy and glucose metabolism, such as pentose phosphate cycle enzymes, CoA biosynthesis and mannose/fructose metabolism, as well as some erythrocyte invasion pathways (*Voorberg-van der Wel et al., 2017*). In schizonts at day 9 this percentage is increased to 93% active pathways (*Figure 2*, *Supplementary file 4*). The only pathways with a decrease in gene activity compared to day 6/7 are involved in remodeling of the host erythrocyte, and expression of genes in the apicoplast. Indeed clustering of the pathways into higher level groups of similar functions (http://mpmp.huji.ac.il/; April, 2018) revealed that cytoadherence is the only function with expression of less than 70% of the genes (*Figure 2—figure supplement 1A*).

In contrast, eight pathways with an increase of over 50% in gene activity at day 9 compared to day 6/7 represent almost exclusively functions involved in parasite invasion, DNA replication and homologous recombination (*Supplementary file 4*). Hence, we conclude that our data at day 9 reflects the maturation of merozoites in late schizogony.

In contrast to schizonts, we previously showed that day 6/7 hypnozoites express less than half of the annotated malaria pathways reflecting the quiescent state and low metabolism that may be expected in dormant forms (*Voorberg-van der Wel et al., 2017*). With only 49 out of 257 (19.1%) pathways expressing more than half of their constituent genes above 1 FPKM, the total number of active pathways is even lower in hypnozoites at day 9 (*Figure 2*).

For further analysis, the 257 pathways were clustered into higher-level groups of similar functions (http://mpmp.huji.ac.il/; April, 2018) and, for each group, the mean percentage of genes active in the pathways was calculated for hypnozoites at day 6/7 and day 9. This revealed lower transcription levels of genes involved in transcription, translation, replication and merozoite invasion in the day 9 hypnozoites. Functions such as chromatin structure and energy metabolism were less represented in hypnozoites at day 9, however, still showed a moderate expression level (*Figure 2—figure supplement 1B*).

A closer look at the energy metabolism in hypnozoites revealed that while glyoxalase, mannose and fructose metabolism are suppressed in hypnozoites at day 9, the pyruvate metabolism, the pentose phosphate cycle and glycolysis are expressed at similar levels as in hypnozoites at day 6/7, suggesting that these pathways represent the main energy source for dormant stages (*Figure 2—figure supplement 1C*). Moreover, histone acetylation and methylation as well as chaperone-mediated modulation of nucleosome-histone interactions show from 50% to 100% activation in day 6/7 and day nine hypnozoites, suggesting a possible role of these pathways in the maintenance of liver stage dormancy (*Figure 2—figure supplement 1C*).

Further, we show that functional processes such as cytoadherence were already expressed at a very low level in the hypnozoites at day 6/7 and stayed at a very low level at day 9. Interestingly, functions such as protein export (e.g. PTEX), chaperones and protein structure modifications (e.g. HSP70 cycle), mitochondrial respiration and redox metabolism showed either upregulation or sustained expression in day 9 hypnozoites, suggesting a key role in the maintenance of the quiescent state (*Figure 2—figure supplement 1B*, *Supplementary file 4*). Moreover, we show that genes required for fatty acid synthesis, genes coding for enzymes involved in post-translational modifications and genes required for the ATP homeostasis are relatively highly expressed in hypnozoites (*Figure 2—figure supplement 1B*, *Supplementary file 4*). The latter is of particular interest, since ATP homeostasis was shown to play an essential role in dormancy in other organisms, for example for the non-replicating dormant form of *Mycobacterium tuberculosis* (*Rao et al., 2008*).

To conclude, our data show that gene expression levels of housekeeping pathways in later stage, day 9 hypnozoites are dampened. Only pathways previously associated with quiescence and required for energy metabolism and maintenance of chromosome integrity remain expressed in hypnozoites.

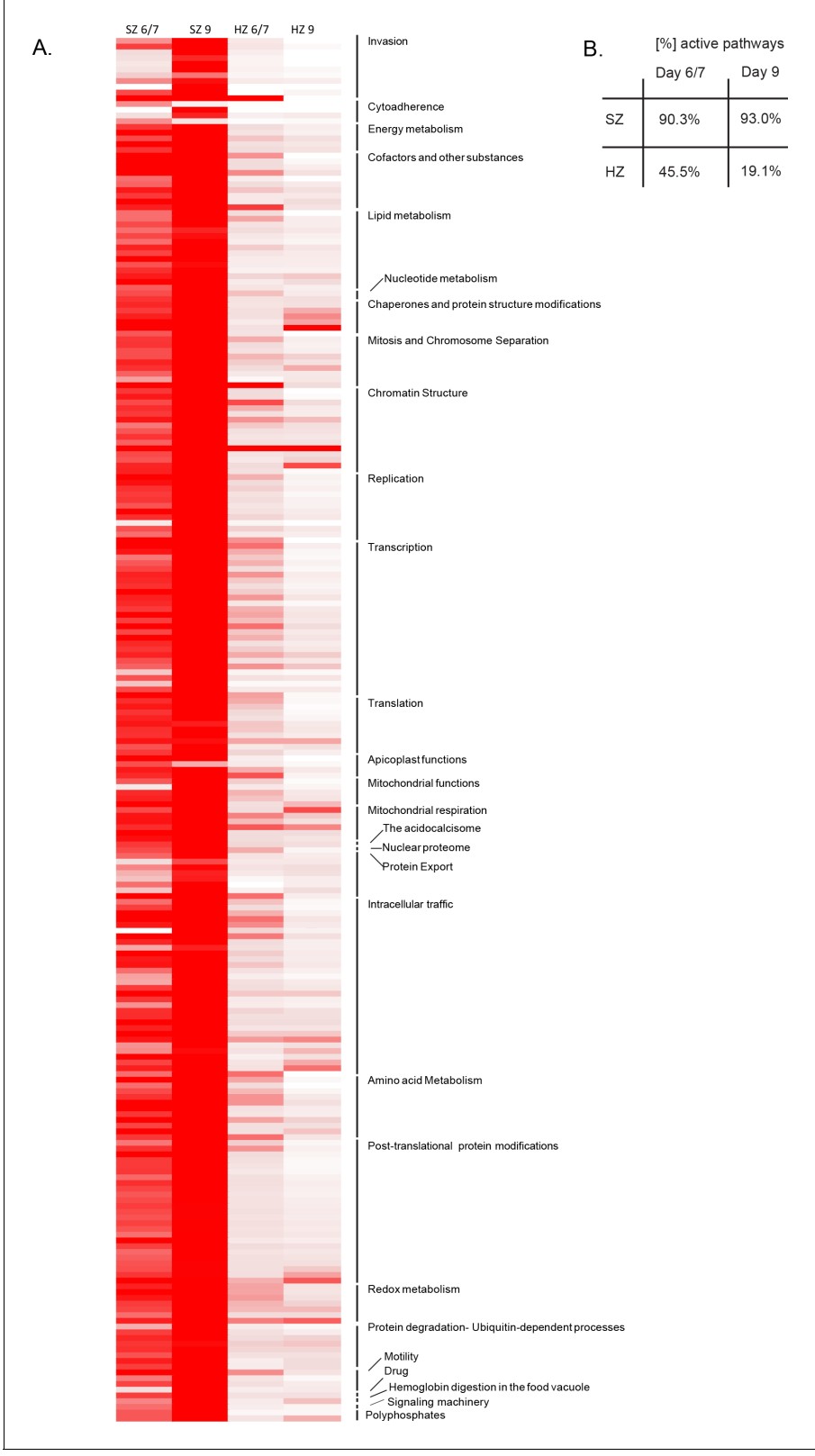

**Figure 2.** Pathway analysis of the malaria liver stages. (**A**) Heat map representing expression of *Plasmodium* pathways in schizonts and hypnozoites at days 6/7 and 9. A total of 257 biological pathways annotated in *P. falciparum* were assigned to *P. cynomolgi* through orthology (see Materials and methods in *Voorberg-van der Wel et al., 2017*). Pathway expression is shown with a color gradient from white (where the fraction of

*Figure 2 continued on next page*

*Figure 2 continued*

genes detected above 1 FPKM is 0%) to red (where this fraction is 100%). (B) Overall percentage of active pathways in schizonts and hypnozoites at days 6/7 and 9.

DOI: https://doi.org/10.7554/eLife.41081.008

The following figure supplement is available for figure 2:

**Figure supplement 1.** Expression levels of selected pathways of interest.

DOI: https://doi.org/10.7554/eLife.41081.009

## Mature hypnozoites express genes involved in the export and transport of parasite proteins into the host-cell

Although dormant, the hypnozoite may still interact with its host cell for survival. Two proteins at the parasite-host interface, the *up-regulated in infective sporozoites gene 4* (UIS4) and the *exported protein-1* (EXP-1), located at the site of the parasitophorous vacuole membrane (PVM), have indeed been shown to interact with host cell proteins in rodent malaria models (*Sá E Cunha et al., 2017*; *Mueller et al., 2005*; *Petersen et al., 2017*). In *P. vivax*, UIS4 was shown to be present in hypnozoites and schizonts throughout liver stage development, while EXP-1 expression was only observed in mid-stage schizonts (*Mikolajczak et al., 2015*). We found transcripts for those genes in both schizonts and hypnozoites in our day 9 transcriptome data and generated *P. cynomolgi* antibodies against these proteins for validation (*Figure 3—figure supplement 1A*). As expected, IFA with UIS4 antibodies showed signal in both hypnozoites and schizonts at day 9 (*Figure 3A*). However, we also observed EXP-1 staining in both parasite forms at day 9. Similar to *P. vivax*, EXP-1 staining was close to background at day 2, while UIS4 staining was clearly visible (*Figure 3—figure supplement 1B*). From day 3/4 onwards EXP-1 staining gradually increased over time and appeared to co-localize with the PVM, both in schizonts and hypnozoites. Further investigation of *P. vivax* liver stage parasites from a humanized mouse model confirmed the presence of EXP-1 in hypnozoites (*Figure 3B*, procedures were used as described in *Mikolajczak et al. (2015)*), indicating that our findings are not an artifact from cell culture. This shows that, while expression of most genes in the hypnozoite is decreased, other genes may be switched on during its maturation into true dormancy.

In the RNA-seq data set of day 9, we found relatively high transcript levels for members of the Alba gene family in hypnozoites, which is in concordance with the previously published data of days 6/7 (*Figure 3—figure supplement 1C*) (*Voorberg-van der Wel et al., 2017*). This gene family plays a crucial role in transcriptional and translational regulation during zygote development as well as in the blood stage of infection (*Reddy et al., 2015*; *Mair et al., 2010*). Indeed we found strong signals for Alba1 transcripts by RNA-FISH at days 6 (not shown) and day 10, both in hypnozoites and schizonts (*Figure 3C*), indicating that this gene may also fulfill important regulatory roles in liver stage development, including hypnozoites.

Further, we showed that protein export is upregulated in hypnozoites at day 9 compared to hypnozoites at day 6/7 (*Figure 2—figure supplement 1B*). This includes four out of the five known components of the *Plasmodium* translocon PTEX (*Figure 3—figure supplement 1C*). This complex composed of HSP101, EXP2, TRX2, PTEX150 and PTEX88 plays an essential role for trafficking exported parasite proteins into the erythrocyte cytoplasm or onto the erythrocyte surface during the blood stage of infection in *P. falciparum* (*Elsworth et al., 2014*). Intriguingly, however, it was shown that most known components of the PTEX translocon are expressed in *Plasmodium* liver stages, corroborating our data (*Nyboer et al., 2018*). RNA-FISH revealed that indeed PTEX150 transcripts were found both in day 6 (not shown) and day 10 schizonts and hypnozoites (*Figure 3C*). This confirms that PTEX is not unique to the blood stage of infection, but also is present in the liver stage of infection, including hypnozoites. However, PTEX functional role in liver-stages remains to be investigated and might differ from that of blood stage of infection (*Vaughan et al., 2012*; *Kalanon et al., 2016*; *Matz et al., 2015*).

Taken together, our data show that hypnozoites display lower transcriptional activity with progressing dormancy. With the markedly reduced transcription levels at day 9 post *in vitro* infection compared to day 6/7, we hypothesize that day 9 hypnozoites are the truly dormant parasites which new drugs need to be active against.

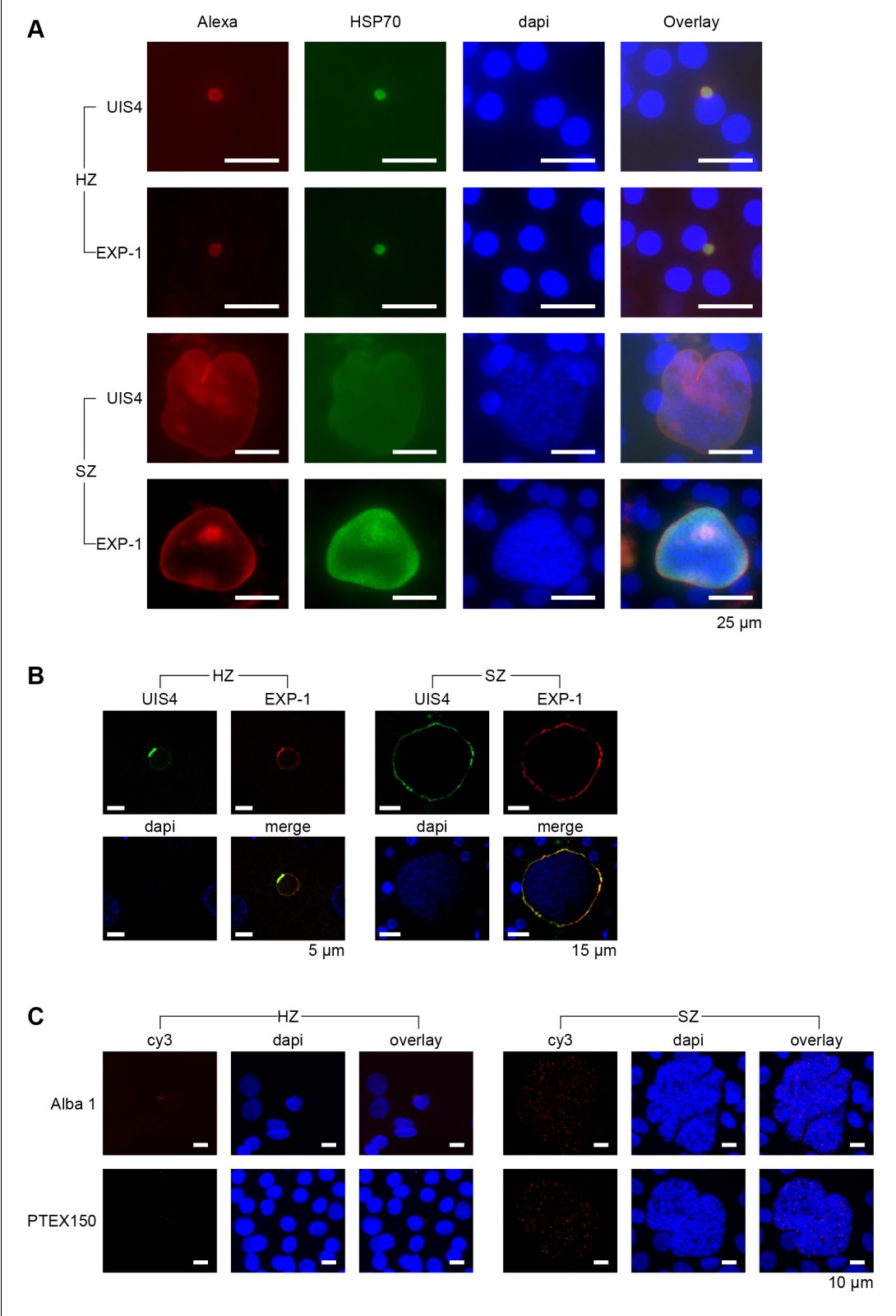

**Figure 3.** Validation of genes involved in the export and transport of parasite proteins. (**A**) Immunofluorescent staining patterns of UIS4 (Alexa-594), EXP-1 (Alexa-594) and HSP70 (FITC) in day 9 *P. cynomolgi* hypnozoites (HZ) and liver schizonts (SZ). Scalebar 25 µm. (**B**) EXP-1 is expressed in *P. vivax* hypnozoites (HZ) and liver schizonts (SZ) at day 8 post sporozoite infection. Colocalization of EXP-1 and UIS4 suggests that EXP-1 is expressed on the
*Figure 3 continued on next page*

*Figure 3 continued*

PVM, in agreement with other *Plasmodium* species. DNA was localized with DAPI. Scalebar 5 µm (HZ) and 15 µm (SZ). (C) RNA-FISH staining of Alba1 (cy3) and PTEX150 (cy3) in day 10 *P. cynomolgi* hypnozoites (Hz) and liver schizonts (Sz). Scalebar 10 µm.

DOI: https://doi.org/10.7554/eLife.41081.010

The following figure supplement is available for figure 3:

**Figure supplement 1.** Validation of selected genes.

DOI: https://doi.org/10.7554/eLife.41081.011

## Discussion

These data add to the previously published transcriptome of day 6/7 parasites (*Voorberg-van der Wel et al., 2017*) and allows for the first time to study liver-stage schizont and hypnozoite maturation. Our analysis shows that maturation of liver schizonts is associated with a general increase in transcriptional activity. This includes genes previously shown to be essential for merozoite formation and red blood cell invasion (*Roth et al., 2018*; *Swann et al., 2016*). The *in vivo* staining patterns with immune-reagents used to characterize liver stage morphologies (*Roth et al., 2018*) confirmed proper liver stage development.

Amongst merozoite-specific genes, day 9 schizonts also revealed upregulation of Pvs16, a gametocyte-specific gene. Together with the data of a recent study, showing that a portion of *P. vivax* late-stage schizonts expresses the Pvs16 protein (*Roth et al., 2018*), this result suggests that some merozoites are already committed as gametocytes prior to the erythrocytic cycle.

In contrast to schizont maturation, we show that progressing dormancy is associated with a metabolic shutdown. Even though transcriptional levels are clearly reduced in day 9 hypnozoites compared to day 6/7 hypnozoites, there are still some specific pathways that show notable expression levels in the mature dormant stage. This includes functions such as energy metabolism, redox metabolism, mitochondrial respiration and chromatin maintenance. Interestingly, these are common features of dormancy found in other microorganisms (*Rittershaus et al., 2013*). Exposed to stress, many microorganisms enter a hardy, non-replicating state, which is able to tolerate adverse environmental conditions, immune responses and prolonged anti-microbial treatments. While cellular adaptations are not exactly the same for all organisms, recent research found some common features of quiescent cells (*Rittershaus et al., 2013*). In the following paragraph, these features are discussed in the light of the acquired data for *Plasmodium* dormant stages.

Our data propose a switch in energy metabolism, suggesting that pyruvate metabolism, the pentose phosphate cycle and glycolysis remain active in the dormant stage, while glyoxalase, mannose and fructose metabolism are suppressed. In other organisms, the switch to a latent lifestyle was indeed shown to be accompanied by a shift in energy metabolism (*Rittershaus et al., 2013*). Similar to our data, in *Toxoplasma gondii* the formation of latent bradyzoites is accompanied by upregulation of glycolysis (*Jeffers et al., 2018*; *Sullivan and Jeffers, 2012*). This is reflected by the prominent expression of pyruvate kinase and lactate dehydrogenase (LDH2) in the latent form as well as confirmed by a knockdown study, showing that LDH2 is essential for bradyzoite formation (*Al-Anouti et al., 2004*). Notably, pyruvate kinase (PcyM_1123400; 3.7 FPKM at day 6/7 (mean) and 1.3 FPKM at day 9 (mean)) and lactate dehydrogenase LDH (PcyM_1234100; 3.7FPKM at day 6/7 (mean) and 3 FPKM at day 9 (mean)) are as well expressed in our transcriptomic dataset of day 6/7 and day 9 hypnozoites, suggesting similar mechanisms.

Moreover, it was shown that *Mycobacteria tuberculosis* redirects acetyl-CoA from the TCA cycle into fatty acid synthesis to build up carbon storage in the latent stage (*Baek et al., 2011*). We show that genes required for fatty acid synthesis are constantly expressed at high levels in hypnozoites. This suggests that hypnozoites, similar to *M. tuberculosis*, accumulate lipids as carbon storage in preparation for long periods of inactivity.

Moreover, in *M. tuberculosis*, ATP homeostasis was shown to be critical for survival. In the latent form of *M. tuberculosis*, ATP level is slightly lower compared to proliferating cells, but it is maintained at a steady state. Depletion of ATP or inhibition of the F0F1 ATP synthase involved in ATP synthesis, results in cell death. This shows that de novo ATP synthesis is required to maintain the low ATP level in *M. tuberculosis*, and hence is critical for dormancy (*Rao et al., 2008*). Intriguingly, genes coding for ATP synthase complex are transcribed in day 6/7 hypnozoites and show upregulation in

day 9 hypnozoites. It is hence tempting to speculate that similar mechanism as found in the dormant form of *Mycobacteria* are critical for dormancy in *Plasmodium* parasites.

In addition, in *Toxoplasma* and *Mycobacteria* it was shown that dormant stages are continually exposed to endogenous and exogenous reactive oxygen species (*Rittershaus et al., 2013*; *Kehrer and Klotz, 2015*). It is hence proposed that latent forms must be capable of dealing with long-term exposure to radicals and reactive metabolic byproducts. Indeed, *T. gondii* and *M. tuberculosis* induce a number of enzymes in the latent stage with roles metabolizing oxygen radicals to maintain balanced oxidation-reduction (*Manger et al., 1998*; *Kumar et al., 2011*). Similar to the data of *Toxoplasma* and *Mycobacteria*, we show that redox metabolism is highly active in day 6/7 hypnozoites and stays active with progressing dormancy. Identification of a mechanism which allows the hypnozoite to counter oxidoreductive stress is central for the development of effective intervention strategies, since its perturbation might either kill or awake the parasite.

Stress response pathways were previously observed to be associated with induction of dormancy. Apart from redox metabolism, we found the heatshock protein 70 (*hsp70*) chaperone cycle to be highly active in late stage hypnozoites. These data were validated by RNA-FISH and immunofluorescence assays. Intriguingly, in *Toxoplasma* expression of *hsp70* is induced during bradyzoite differentiation and its inhibition can suppress bradyzoite development *in vitro* (*Weiss et al., 1998*). Since members of the HSP70 family are associated with stage transition in many different organisms, it is tempting to speculate that, similar to *Toxoplasma*, HSP70 plays a crucial role in the formation and maintenance of the dormant stage. Identification of interaction partners of HSP70 may yield information about the process of stage transition.

Interestingly, a recent study uncovered a complex ApiAP2 transcriptional network of repressors and activators controlling the switch between replicating tachyzoite and latent bradyzoite formation in *Toxoplasma gondii* (*Lesage et al., 2018*). The ApiAP2 family includes plant-like transcription factors that are key regulators of life cycle transition in *Apicomplexan* parasites. It is speculated that the AP2VIIa-1 is a master regulator which coordinates changes in the *Toxoplasma* transcriptome that leads to bradyzoite conversion (*Radke et al., 2018*). In *Plasmodium spp.* several ApiAP2 transcription factors were identified to play a crucial role in gametocytogenesis (*Kafsack et al., 2014*; *Sinha et al., 2014*; *Yuda et al., 2015*), ookinete development (*Yuda et al., 2009*), sporozoite formation (*Yuda et al., 2010*) and liver stage maturation (*Iwanaga et al., 2012*). So far, however, it remains elusive if ApiAP2 factors regulate hypnozoite formation, maturation and/or reactivation. Here, we show that *ap2-l*, an essential factor for liver stage schizont maturation (*Iwanaga et al., 2012*), is not only expressed in schizonts, but also in hypnozoites, suggesting an important role as well in hypnozoite development. In contrast, we could not confirm expression of the proposed quiescence ApiAP2 factor identified by (*Cubi et al., 2017*). Instead, we show that the ApiAP2 transcription factor PcyM_0918000 is transcribed in hypnozoites, corroborating the results of a recent transcriptome study in *P. vivax* (*Gural et al., 2018*). However, this factor is not exclusive for the dormant stage as suggested for a master regulator for hypnozoite conversion. It hence remains to be determined if indeed this ApiAP2 transcription factor and/or an ApiAP2 transcriptional network play a key role in regulating the hypnozoite fate.

Apart from ApiAP2 transcription factors as master regulators of hypnozoite fate, it is speculated that epigenetic control might be implicated in the maintenance of quiescence of hypnozoites (*Malmquist et al., 2012*). Indeed, exposure of dormant stages to an inhibitor of histone lysine methyltransferases induced an accelerated rate of hypnozoite activation (*Dembélé et al., 2014*). It is hence tempting to speculate that epigenetic processes such as methylation-dependent changes in transcription might control hypnozoite quiescence. Indeed, our data show that pathways of histone acetylation and methylation as well as chaperone-mediated modulation of nucleosome-histone interactions are highly expressed in days 6/7 and day 9 hypnozoites, suggesting a key role in maintaining dormancy.

To maintain genome fidelity is challenging for quiescent cells, since the low metabolic activity allows for only limited DNA repair mechanisms. One common strategy is to alter chromosomal structure to a more chemically stable form. In *M. tuberculosis* a histone-like protein Lsr2, mediates chromosome compaction and protection from reactive oxygen and nitrogen species (*Summers et al., 2012*). In our transcriptomic dataset, we found that the *P. cynomolgi* homologue of the bacterial histone-like protein (HU) (PcyM_0702400) is expressed in hypnozoites (3.5 FPKM at day 6/7 and 2.2 FPKM at day 9). It is hence tempting to speculate that this protein might play a role in chromosome

condensation during dormancy. Moreover, we find Alba genes transcribed in hypnozoites. Members of the Alba family from protozoan parasites bind to both DNA and RNA and are implicated in transcriptional regulation, chromatin packaging, and translational control, as well as cellular differentiation and developmental processes (*Goyal et al., 2016*). In *P. falciparum* for example, Alba proteins were shown not only to drive translational repression in sporozoites, but also to be associated with subtelomeric DNA and epigenetic regulators, suggesting a role in heterochromatin formation (*Vembar et al., 2015*; *Goyal et al., 2012*). Further studies are needed to map epigenetic modifications in hypnozoites, which will hopefully shed light on the complex developmental pathway driving dormancy.

Finally, we show the translocon PTEX is actively transcribed in hypnozoites. So far, this complex was only described as essential for protein export in blood stage parasites (*Elsworth et al., 2014*). Whether this complex is indeed also important for nutrient acquisition in liver stage parasites, including hypnozoites, remains to be determined.

To conclude, although we do not identify a specific transcriptional marker for hypnozoites because dormancy is likely associated with a general metabolic shutdown, there are still several specific pathways that show notable expression levels, at least at the transcriptional level, in the mature dormant stage. Interestingly, these pathways were shown to be essential for dormancy in other organisms. Therefore specific targeting of these functions may be a productive strategy to identify novel effective therapies.

# Materials and methods

## Ethics statement

Nonhuman primates were used because no other models (*in vitro* or *in vivo*) were suitable for the aims of this project. The research protocol was approved by the local independent ethical committee conform Dutch law (BPRC Dier Experimenten Commissie, DEC, agreement number #708). Details are described by *Voorberg-van der Wel et al. (2017)*.

## Transgenic *Plasmodium cynomolgi* sporozoite production

*P. cynomolgi* M strain PcyC-PAC-GFP$_{hsp70}$-mCherry$_{ef1\alpha}$ (*Voorberg-van der Wel et al., 2013*) sporozoites were produced as described previously (*Voorberg-van der Wel et al., 2017*).

## Parasite liver stage culture and cell sorting

Procedures for liver cell isolation, liver stage culture and cell sorting were essentially as described previously (*Voorberg-van der Wel et al., 2017*). Briefly, transgenic *P. cynomolgi* salivary gland sporozoites were isolated and added to freshly isolated *Macaca mulatta* hepatocytes. Hepatocytes were seeded in 96-well collagen-coated plates at 90,000 hepatocytes per well and 2 days later 50,000 sporozoites were added per well. After 9 or 10 days of culture, infected cells were sorted with a BD FACSAria flowcytometer and fractions were collected in Trizol and stored at −80°C until RNA extraction. Median fluorescence intensity (MFI) values of GFPlow and GFPhigh samples were calculated using four separate recordings of 1 million cells per experiment (day 6, n = 4; days 9 and 10, n = 3).

## Protein and antibody production

PcyM_0602100 (UIS4) aa20-166 and PcyM_0924700 (EXP-1) aa19-130 were expressed in *E. coli*, purified using a Ni-IMAC column followed by gel-filtration and used to immunize rats (Eurogentec, Belgium).

## Immunofluorescence analysis (IFA)

IFA staining of methanol fixed liver stage parasites was carried out as before (*Voorberg-van der Wel et al., 2017*). Alternatively, parasites were fixed in 4% paraformaldehyde (30 min., room temperature), washed and incubated with primary and secondary antibodies diluted in 1% BSA/0.3% Triton X-100 in PBS.

## RNAscope in situ hybridization

For RNA-FISH, *P. cynomolgi* liver stage cultures in 96-well collagen-coated CellCarrier plates (PerkinElmer) were washed once in PBS and fixed in 4% paraformaldehyde (Affymetrix) at room temperature for 30 min. Cultures were then washed, dehydrated and stored in 100% ethanol at −20°C until RNA-FISH was performed using a Tyramide Signal Amplification (TSA) based assay from Advanced Cell Diagnostics (RNAscope Multiplex Fluorescent Assay v2), essentially according to the manufacturer's instructions. Following rehydration protease digestions were performed using pretreatment solution three from the kit at 1:10 dilution for 20 min. at room temperature. Hybridizations were 2 hr at 40°C. Probes used were directed against *P. cynomolgi* hsp70 (PcyM_0515400, region 606–1837 of XM_004221103.1), PTEX150 (PcyM_1315200 targeting 871–2360 of XM_004224250.1), Alba1 (PcyM_1427300; targeting 101–692 of PcyM_1427300) and MSP1 (PcyM_0731200 targeting 147–1127 of XM_004221774.1). After hybridization, TSA amplification steps were performed as described by the manufacturer. Following DAPI staining, cells were kept in PBS for imaging. Images were acquired with a Leica DMI6000B inverted fluorescence microscope equipped with a DFC365FX camera using a HC PL APO 63x/1.40–0.60 oil objective.

## RNA sequencing

Total RNA was isolated using the Direct-zol RNA MiniPrep Kit (Zymo Research) including on-column DNase digestion according to the manufacturer's instructions. The quality of the RNA samples was assessed with the High Sensitivity RNA kit using the TapeStation 4200 instrument (Agilent Technologies).

RNA samples were processed using the SMART-Seq v4 Ultra Low Input RNA Kit (Clontech) to generate high-quality cDNA. The results obtained were evaluated with the High Sensitivity DNA kit using the Bioanalyzer 2100 instrument (Agilent Technologies). The cDNA samples were then sheared to 200–500 bp length using a Covaris E220 instrument (Covaris) and subsequently processed with the Low Input Library Prep Kit v2 (Clontech) to generate sequencing libraries. The quality of the libraries was assessed with the D1000 TapeStation kit (Agilent Technologies). RNA-seq cDNA libraries were sequenced in paired-end mode, 2 × 76-base-pair (bp), using the Illumina HiSeq2500 platform. Read quality was assessed by running FastQC (version 0.10) on the FASTQ files. Sequencing reads showed high quality, with a mean Phred score higher than 30 for all base positions. The obtained 76-bp paired-end reads were used for the alignment to the reference genomes and used for the gene expression quantification with the Exon Quantification Pipeline (EQP) (*Schuierer and Roma, 2016*) as described in *Voorberg-van der Wel et al. (2017)*.

To quantify parasite-specific expression for each *P. cynomolgi* gene, we determined the number of sequencing reads aligned to genes and computed gene expression values as the number of Fragments per Kilobase per Million fragments mapped (FPKM). For the normalization of the schizont expression values, we applied the previously developed method of normalization against the total number of host reads (*Voorberg-van der Wel et al., 2017*). For the hypnozoite expression values, we used a different normalization scheme since we detected uninfected host cells in the day 9/10 hypnozoite samples via microscopy. These contaminating uninfected host cells led to a higher content of monkey reads in the day 9/10 hypnozoite samples as compared to the day 6/7 hypnozoite samples which were described in *Voorberg-van der Wel et al. (2017)*. So after discarding the samples of day 10 we decided to completely disregard the percentage of monkey reads in the hypnozoite samples and instead use a standard FPKM normalization which only takes the number of parasite reads into account (*Love et al., 2014*). Since we only compare schizont samples with each other and hypnozoite samples with each other, using two different normalization schemes still leads to consistent results. All data are based on FPKM values after such normalization which are provided in supplementary (*Supplementary file 5*).

## Acknowledgements

We are grateful to the mosquito breeding facilities in Nijmegen for provision of *Anopheles stephensi* mosquitoes. We thank Francisca van Hassel for preparing graphical representations. We are grateful to the Kappe and Sattabongkot laboratories for continuous support and collaboration in the area of *P. vivax* liver stage research.

# Additional information

## Competing interests

Nicole L Bertschi, Sven Schuierer, Florian Nigsch, Walter Carbone, Judith Knehr, Devendra K Gupta, Erika L Flannery, Sebastian A Mikolajczak, Vorada Chuenchob, Binesh Shrestha, Martin Beibel, Tewis Bouwmeester, Thierry T Diagana, Guglielmo Roma: Employed by and/or shareholder of Novartis Pharma AG. The other authors declare that no competing interests exist.

## Funding

| Funder | Author |
| --- | --- |
| Bill and Melinda Gates Foundation | Thierry T Diagana<br>Clemens HM Kocken<br>Guglielmo Roma |
| Wellcome | Thierry T Diagana<br>Clemens HM Kocken |
| Medicines for Malaria Venture | Thierry T Diagana<br>Clemens HM Kocken |

The funders had no role in study design, data collection and interpretation, or the decision to submit the work for publication.

## Author contributions

Nicole L Bertschi, Formal analysis, Investigation, Visualization, Writing—original draft, Project administration, Writing—review and editing; Annemarie Voorberg-van der Wel, Formal analysis, Validation, Investigation, Visualization, Methodology, Writing—original draft, Project administration, Writing—review and editing; Anne-Marie Zeeman, Walter Carbone, Judith Knehr, Nicole van der Werff, Ivonne Nieuwenhuis, Els Klooster, Binesh Shrestha, Formal analysis, Investigation, Methodology; Sven Schuierer, Florian Nigsch, Software, Formal analysis, Investigation, Methodology; Devendra K Gupta, Formal analysis, Validation, Investigation, Writing—review and editing; Sam O Hofman, Vorada Chuenchob, Validation, Formal analysis, Methodology; Bart W Faber, Formal analysis, Methodology; Erika L Flannery, Sebastian A Mikolajczak, Validation, Investigation, Writing—review and editing; Martin Beibel, Software, Formal analysis, Methodology, Writing—review and editing; Tewis Bouwmeester, Resources, Investigation, Writing—review and editing; Niwat Kangwanrangsan, Jetsumon Sattabongkot, Resources, Validation, Investigation; Thierry T Diagana, Conceptualization, Resources, Formal analysis, Supervision, Funding acquisition, Investigation, Project administration, Writing—review and editing; Clemens HM Kocken, Conceptualization, Resources, Data curation, Formal analysis, Supervision, Funding acquisition, Investigation, Methodology, Writing—original draft, Project administration, Writing—review and editing; Guglielmo Roma, Conceptualization, Resources, Data curation, Formal analysis, Supervision, Funding acquisition, Investigation, Visualization, Methodology, Writing—original draft, Project administration, Writing—review and editing

## Author ORCIDs

Annemarie Voorberg-van der Wel (iD) http://orcid.org/0000-0001-9403-0515
Walter Carbone (iD) https://orcid.org/0000-0001-6150-8295
Erika L Flannery (iD) https://orcid.org/0000-0003-0665-7954
Jetsumon Sattabongkot (iD) https://orcid.org/0000-0002-3938-4588
Thierry T Diagana (iD) https://orcid.org/0000-0002-8520-5683
Guglielmo Roma (iD) https://orcid.org/0000-0002-8020-4219

## Ethics

Animal experimentation: Nonhuman primates were used because no other models (in vitro or in vivo) were suitable for the aims of this project. The research protocol was approved by the local independent ethical committee conform Dutch law (BPRC Dier Experimenten Commissie, DEC, agreement number #708). Details are described by (Voorberg-van der Wel et al., 2017).

Decision letter and Author response
Decision letter https://doi.org/10.7554/eLife.41081.024
Author response https://doi.org/10.7554/eLife.41081.025

## Additional files

### Supplementary files

• Supplementary file 1. *P.cynomolgi* samples used for RNA-seq, *related to Figure 1*.
DOI: https://doi.org/10.7554/eLife.41081.012

• Supplementary file 2. Genes with expression vaues $\geq$ 3 FPKM in hypnozoite samples at day 9 or day 10, *related to Figure 1—figure supplement 3B*.
DOI: https://doi.org/10.7554/eLife.41081.013

• Supplementary file 3. Genes with expression vaues $\geq$10 FPKM in hypnozoite samples at day 9 and/or day 10, *related to Figure 1*.
DOI: https://doi.org/10.7554/eLife.41081.014

• Supplementary file 4. Expression of *Plasmodium* pathways in schizont and hypnozoite samples at day 9 and day 10. The table indicates if pathways are significantly up- or down-regulated over time.
DOI: https://doi.org/10.7554/eLife.41081.015

• Supplementary file 5. Gene expression values (FPKM) generated by RNA-seq of hypnozoite (Hz) and liver schizont (Sz) samples, *related to Figure 1*.
DOI: https://doi.org/10.7554/eLife.41081.016

• Transparent reporting form
DOI: https://doi.org/10.7554/eLife.41081.017

### Data availability

The raw RNA-sequencing reads are available in the NCBI Short Read Archive (https://www.ncbi.nlm.nih.gov/sra) under accession number SRP096160.

The following dataset was generated:

| Author(s) | Year | Dataset title | Dataset URL | Database and Identifier |
|---|---|---|---|---|
| Guglielmo Roma | 2018 | Malaria Liver Stages Transcriptome, '18 | https://www.ncbi.nlm.nih.gov/sra/?term=SRP096160 | NCBI Short Read Archive, SRP096160 |

The following previously published dataset was used:

| Author(s) | Year | Dataset title | Dataset URL | Database and Identifier |
|---|---|---|---|---|
| Guglielmo Roma | 2017 | Malaria Liver Stages Transcriptome, '17 | https://www.ncbi.nlm.nih.gov/sra/?term=SRP096160 | NCBI Short Read Archive, SRP096160 |

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
