## [Decision Letter]

[Editors’ note: the authors were asked to provide a plan for revisions before the editors issued a final decision. What follows is the editors’ letter requesting such plan.]

Thank you for sending your article entitled "Transcriptomic analysis reveals malaria hypnozoite maturation" for peer review at *eLife*. Your article is being reviewed by Anna Akhmanova as the Senior Editor Editor, a Reviewing Editor, and three reviewers.

Given the list of essential revisions, including new experiments, the editors and reviewers invite you to respond within the next two weeks with an action plan and timetable for the completion of the additional work. We plan to share your responses with the reviewers and then issue a binding recommendation.

As you can see from the comments included below, each reviewer expressed positive views regarding the organization and the performance of the experiments as well as the robustness of your results on transcriptomics during the development of *P. cynomolgi*. While they considered the information to be of value and worthy of publication, they also commented on the small incremental step that your data provide and thus questioned the advancement towards understanding dormancy of hypnozoites. To substantially enhance your story, they suggested performing additional experiments: (1) to demonstrate protein expression of the genes transcribed during previously reported biomarkers for day 6/7 hypnozoites that were not found at day 9; (2) to validate some pathways revealed at the transcriptomic level.

*Reviewer #1:*

The paper by Bertschi et al. describes the transcriptomes of *P. cynomolgi* liver stage hypnozoites and liver stage schizonts at day 9 and day 10 post infection. The parasite material was derived from *in vitro*-cultured hepatocyte infections that were subject to cell sorting for enrichment of parasite-infected hepatocytes. The work is an extension of data recently published by the same group in *eLife* (Voorberg-van der Wel et al., 2017), which described hypnozoite and schizont transcriptomes from parasites at days 6/7 post infection.

The experimental work is technically sound, and the data analysis is reasonably well done. The main conclusion drawn by the authors is that more 'mature' day 9 hypnozoites exhibit similar gene expression patterns when compared to day 6/7 hypnozoites but the former shows a further decrease of transcription.

While the work is of value and a descriptive and incremental step forward towards the delineation of liver stage gene expression profiles and gene expression in hypnozoites, I cannot see that it provides a significant step towards an understanding of the molecular mechanisms of dormancy and persistence of hypnozoites. It is entirely possible that the answer to the question "what regulates dormancy" will not be revealed by further transcriptome analysis beyond what has been recently published (Voorberg-van der Wel et al., 2017; Gural et al., 2018) and this manuscript further confirms this problem. It might be time that the focus is shifted towards potential posttranscriptional or posttranslational mechanisms that might regulate dormancy.

*Reviewer #2:*

Bertschi and colleagues unravel the transcriptome of 9 to 10-day-old *P. cynomolgi* liver stage parasites, including the maturation of its dormant stage hypnozoite in their manuscript, titled: "Transcriptomic analysis reveals malaria hypnozoite maturation." Data analysis shows that the progression of parasite dormancy leads to a 10-fold decrease in transcription with only 840 transcribed genes. Expression analysis shows that pathways involved in quiescence, energy metabolism and maintenance of genome integrity remain the prevalent pathways active in mature hypnozoites. The manuscript is well written. The data generated are robust, well validated and further confirm the importance of epigenetic control in parasite development. I would, however, like to stress that while the described work has been thoroughly performed, the work appears narrow and incremental to previous published datasets. Additional experiments targeting/validating the identified pathways as essential for hypnozoite persistence would hold more value. Is there anyway the team can test the inhibitory activity of drugs/molecules targeting histone-modifying enzymes such as histone acetyltransferases (HATs), histone methyltransferases (HMT) or histone chaperones against hypnozoite maturation and survival? This would significantly improve the relevance of this manuscript.

*Reviewer #3:*

Although the sample size is small, this manuscript shows convincingly that as hypnozoites mature from day 6/7 to day 9, transcriptional activity is massively reduced. This is in stark contrast to the increased transcriptional activity at the same timepoints during schizont development. There is a strong rationale for doing this work because in order for therapeutics to provide effective radical cure, they must target processes in even the most dormant of hypnozoites. Although the inferences of this study are limited to *in vitro* experiments using non-human malaria, the authors point out several consistencies of these results with *P. vivax* studies and these results also corroborate the prophylactic but not radical curative activity of PI4K. The manuscript is well organized, and the conclusions are supported by the design and analysis of their experiments. I would recommend with work for publication.

[Editors’ note: formal revisions were requested, following approval of the authors’ plan of action.]

Thank you for choosing to send your work entitled "Transcriptomic analysis reveals malaria hypnozoite maturation" for consideration at *eLife*. Your plan has been considered by a Senior Editor, Reviewing Editor and three reviewers, and we are prepared to consider a revised submission.

We will not request any additional experiments, but we will suggest that the authors do respond to the other comments posted by the reviewers – these are mainly editorial in nature – and to revise the title of the manuscript.

---

## [Author Response]

[Editors’ note: what follows is the authors’ plan to address the revisions.]

As you can see from the comments included below, each reviewer expressed positive views regarding the organization and the performance of the experiments as well as the robustness of your results on transcriptomics during the development of P. cynomolgi. While they considered the information to be of value and worthy of publication, they also commented on the small incremental step that your data provide and thus questioned the advancement towards understanding dormancy of hypnozoites. To substantially enhance your story, they suggested performing additional experiments: (1) to demonstrate protein expression of the genes transcribed during previously reported biomarkers for day 6/7 hypnozoites that were not found at day 9;

This is an interesting suggestion; however, the analysis of the day 6/7 transcriptome data unfortunately did not detect any hypnozoite-specific biomarkers. This means that we do not have any hypnozoite biomarkers we could select for the proposed experiment at day 9.

In the previous manuscript, we indicated a dozen of transcripts upregulated in the hypnozoites versus schizonts, which correspond to genes with low expression levels in the hypnozoites and very-low expression levels in the schizonts.Thefollow-up validation of the most upregulated gene in hypnozoites, a member of the ETRAMP family, could not confirm protein expression in sporozoites, hypnozoites or schizonts during liver stage development; this appeared to be due to a different alternative splicing (intron retention events) occurring in these stages, resulting in the presence of premature stop codons in the mRNA sequence and hence in the absence of protein expression at days 6 and 7.Thus, the ETRAMP protein would not qualify as a hypnozoite-specific marker to be validated at day 9.

Selecting the other low-level transcribed genes from the day 6/7 hypnozoite transcriptome, raising antibodies to recombinant proteins (in our hands, antibodies against synthetic peptides are rarely reactive with malaria proteins), preparing new EEF cultures and probing those with the antibodies could easily take 6 months and would require another monkey infection.

In addition, we understand that the reviewer raised this point merely as an interesting route to follow, beyond the scope of our current work, and they do not seem to suggest additional experiments.

(2) to validate some pathways revealed at the transcriptomic level.

Reviewer 2 suggests evaluating the effect of compounds targeting histone-modifying enzymes. This is also an interesting suggestion but would require a massive number of studies that we feel are beyond the scope of this paper. In fact, we would expect the effects of these compounds to be pleiotropic by definition and to lead to the generation of data that may be very difficult to interpret and would require substantial validation even if we see an effect. To more effectively test this type of compounds we would have to develop new types of assays where we can follow individual parasites over time. A pilot experiment with a limited set of inhibitors may likely take at least 6 months, without any guaranteed success. For this reason, we would prefer to consider the drug screening as a new chapter compared to the proposed “research advance” article that builds on and complements our recent paper of malaria liver stage transcriptomics.

[Editors’ notes: the authors’ response after being formally invited to submit a revised submission follows.]

We will not request any additional experiments, but we will suggest that the authors do respond to the other comments posted by the reviewers – these are mainly editorial in nature – and to revise the title of the manuscript.

We carefully addressed the concerns of the referees and give detailed answers to the different points below. In addition, we revised the title of our manuscript as follows:

“Transcriptomic analysis reveals reduced transcriptional activity in malaria hypnozoites during progression into dormancy”.

Reviewer #1:The paper by Bertschi et al. describes the transcriptomes of P. cynomolgi liver stage hypnozoites and liver stage schizonts at day 9 and day 10 post infection. The parasite material was derived from *in vitro*-cultured hepatocyte infections that were subject to cell sorting for enrichment of parasite-infected hepatocytes. The work is an extension of data recently published by the same group in eLife (Voorberg-van der Wel et al., 2017), which described hypnozoite and schizont transcriptomes from parasites at days 6/7 post infection.The experimental work is technically sound, and the data analysis is reasonably well done. The main conclusion drawn by the authors is that more 'mature' day 9 hypnozoites exhibit similar gene expression patterns when compared to day 6/7 hypnozoites but the former shows a further decrease of transcription.While the work is of value and a descriptive and incremental step forward towards the delineation of liver stage gene expression profiles and gene expression in hypnozoites, I cannot see that it provides a significant step towards an understanding of the molecular mechanisms of dormancy and persistence of hypnozoites. It is entirely possible that the answer to the question "what regulates dormancy" will not be revealed by further transcriptome analysis beyond what has been recently published (Voorberg-van der Wel et al., 2017; Gural et al., 2018) and this manuscript further confirms this problem. It might be time that the focus is shifted towards potential posttranscriptional or posttranslational mechanisms that might regulate dormancy.

We thank the reviewer for this thorough review and agree with the need to shift the focus of future studies towards posttranscriptional or posttranslational mechanisms that might regulate dormancy in the hypnozoite.

Reviewer #2:Bertschi and colleagues unravel the transcriptome of 9 to 10-day-old P. cynomolgi liver stage parasites, including the maturation of its dormant stage hypnozoite in their manuscript, titled: "Transcriptomic analysis reveals malaria hypnozoite maturation." Data analysis shows that the progression of parasite dormancy leads to a 10-fold decrease in transcription with only 840 transcribed genes. Expression analysis shows that pathways involved in quiescence, energy metabolism and maintenance of genome integrity remain the prevalent pathways active in mature hypnozoites. The manuscript is well written. The data generated are robust, well validated and further confirm the importance of epigenetic control in parasite development. I would, however, like to stress that while the described work has been thoroughly performed, the work appears narrow and incremental to previous published data sets. Additional experiments targeting/validating the identified pathways as essential for hypnozoite persistence would hold more value. Is there anyway the team can test the inhibitory activity of drugs/molecules targeting histone-modifying enzymes such as histone acetyltransferases (HATs), histone methyltransferases (HMT) or histone chaperones against hypnozoite maturation and survival? This would significantly improve the relevance of this manuscript.

We thank the reviewer for this thorough review and for the suggestion to perform additional drug screening with drugs targeting histone-modifying enzymes. In our revision plan, we have clearly addressed the main challenges related to the execution of these follow up experiments.